



# Technical Note: Data assimilation and autoregression for using near-real-time streamflow observations in long short-term memory networks

Grey S. Nearing[1,2], Daniel Klotz[3], Alden Keefe Sampson[4], Frederik Kratzert[5], Martin Gauch[3], Jonathan M. Frame[6,7], Guy Shalev[8], and Sella Nevo[8]

[1]Google Research, Mountain View, CA, United States
[2]University of California Davis, Department of Land, Air & Water Resources, Davis, CA, United States
[3]LIT AI Lab & Institute for Machine Learning, Johannes Kepler University, Linz, Austria
[4]Upstream Tech, Alameda, CA, USA
[5]Google Research, Vienna, Austria
[6]National Water Center, National Oceanic and Atmospheric Administration, Tuscaloosa, AL, United States
[7]Department of Geological Sciences, University of Alabama, Tuscaloosa, AL, USA
[8]Google Research, Tel Aviv, Israel

**Correspondence:** Grey Nearing (gsnearing@google.com)

**Abstract.** Ingesting near-real-time observation data is a critical component of many operational hydrological forecasting systems. In this paper we compare two strategies for ingesting near-real-time streamflow observations into Long Short-Term Memory (LSTM) rainfall-runoff models: autoregression (a forward method) and variational data assimilation. Autoregression is both more accurate and more computationally efficient than data assimilation. Autoregression is sensitive to missing data, however an appropriate (and simple) training strategy mitigates this problem.

## 1 Introduction

Long Short-Term Memory networks (LSTMs) are currently the most accurate and extrapolatable streamflow models available from the hydrological science community (e.g., Kratzert et al., 2019c, b; Gauch et al., 2021a; Frame et al., 2021). Achieving the highest accuracy simulations possible in an operational setting requires the ability to leverage near-real-time streamflow observation data during prediction, wherever and whenever such data are available. There are two primary ways that rainfall-

10 runoff models use near-real-time streamflow observation data: autoregression and data assimilation.

Autoregression (AR) has been a core component of statistical hydrology for decades (e.g., Matalas, 1967; Fernandez and Salas, 1986; Hsu et al., 1995; Abrahart and See, 2000; Wunsch et al., 2021). AR is also common in machine learning applications across many different types of domain applications (e.g., Uria et al., 2013; Vaswani et al., 2017; De Fauw et al., 2019;

Child, 2020; Salinas et al., 2020; Dhariwal et al., 2020), including LSTM based approaches (e.g., Graves, 2013; Gregor et al., 2015; Van Oord et al., 2016). Most importantly for this discussion, Feng et al. (2020); Moshe et al. (2020) showed that AR improves streamflow predictions from LSTMs. AR modeling with LSTMs is complicated somewhat by the fact that LSTMs are sensitive to missing data. In particular, a naive LSTM simply cannot run if any of its inputs are missing, and missing near-



real-time data is an issue for operational forecasting. It is possible to mitigate the problem of missing data in LSTM inputs by
masking (e.g., Chollet, 2017, chapter 4), gap filling, or adversarial learning to impute missing data (Kim et al., 2020; Dong
et al., 2021), however these strategies necessarily introduce some amount of bias in the inputs.

In contrast with statistical autoregressive models, conceptual and process-based rainfall-runoff models typically use data as-
similation (DA) to ingest near-real-time streamflow observations. There are a number of different DA methods used in the Earth
sciences (Reichle, 2008), ranging from direct insertion to full Markov Chain Monte Carlo approximations of nonlinear, non-
Gaussian conditional probabilities (e.g., van Leeuwen, 2010). Most DA methods use filters or smoothers based on simplified
probabilities – for example, variations of Kalman-type filters minimize variance (e.g., Evensen, 2003), particle filters maxi-
mize more general likelihoods (e.g., Del Moral, 1997) but run into challenges related to high dimensional sampling (Snyder
et al., 2008), and variational filters numerically minimize specified loss functions (Rabier and Liu, 2003). All data assimilation
methods fundamentally work by conditioning (changing) the states of a dynamical systems model so that information from
observations persist in the model for some amount of time.

Like dynamical systems models, LSTMs have a recurrent state. This means that it is possible to use DA with LSTMs. This
would allow ingesting near-real-time observation data without AR, making it possible to train LSTM models that are able to
leverage near-real-time streamflow data where and when available. Further, LSTMs are trained with backpropagation, which
means that there already exists a gradient chain through the model's tensor network that can be used for implementing certain
types of inverse methods required for DA. Similar principles have been applied to update other features in deep learning models
for a variety of purposes. For example, backpropagation to update inputs and specific layers has been used as an analytical tool
(e.g., Olah et al., 2017; Dosovitskiy and Brox, 2016; Mahendran and Vedaldi, 2015) and to generate adversarial examples for
training (Szegedy et al., 2013).

The major concern with statistical approaches (like AR) is that they often do not generalize to new locations or to situations
that are dissimilar to the training data (e.g., Cameron et al., 2002; Gaume and Gosset, 2003). However, rainfall-runoff models
based on LSTMs generalize to ungauged basins Kratzert et al. (2019b) and extreme events (Frame et al., 2021) better than both
conceptual and process-based hydrology models. The performance of LSTM-based rainfall-runoff models can be improved
significantly using autoregression, however LSTMs are sensitive to missing data, which means that ad hoc approaches must
be employed for gap-filling. Additionally, autoregressive models cannot be applied to ungauged catchments where lagged
streamflow data are not available. Missing streamflow data is a common problem in operational settings, since in many parts
of the world streamflow data are collected by hand or using sensors that are prone to malfunction, large measurement error, or
breaks in communication with data loggers. As an example, the Google flood forecasting model (Nevo et al., 2019) typically
faces roughly 10% - 30% missing streamflow data during each monsoon season in different watersheds in India. DA has at
least a potential advantage over AR in that it is robust to missing data: whenever there is no observation data to assimilate, the
original model continues to make predictions.

The purpose of this paper is to provide insight into trade-offs between DA and AR for leveraging potentially sparse near-
real-time streamflow observation data. AR is easier to implement than DA (simply train a model with autoregressive inputs),
and it is also more computationally efficient because it does not require any type of inverse procedure during prediction (e.g.,





variational optimization, ensembles for estimating conditional probabilities, high dimensional particle sampling, etc.). Inverse
procedures used for DA not only require significant computational expense, but also are sensitive to hyperparameters related
to things like error distributions, regularization coefficients, and resampling procedures (Nearing et al., 2018; Bannister, 2017;
Snyder et al., 2008). On the other hand, AR suffers from effects of missing data, which is a serious problem in an operational
setting. In this paper we compare (i) a very simple procedure for dealing with missing data data in an AR LSTM model with
(ii) variational data assimilation (also applied to an LSTM model), and show that the simple AR approach works better.

As a caveat, it is important to point out that there are many strategies that probably could be developed or employed to deal
with missing input data in LSTM streamflow models. Additionally, there are many different types of DA methods that could
be used (e.g., EnKF, particle filters, etc.). The variational DA strategy that we test here is one of the most common forms of
DA, and one that is easily implemented in an ML setting because it directly leverages the deep learning tensor network. It
is impossible for us to test every possible method for AR and DA that has been developed, however the goal of this paper is
65 to provide guidance to forecasters and model developers about which of these approaches are worth pursuing in operational
models. Our suggestion based on results presented within is that, at present, we suggest that it is likely more promising to
develop AR-based approaches rather than DA-based approaches.

## 2   Methods

### 2.1   Data

To allow for direct comparison with previous studies, we tested autoregression and backpropagation-based variational data
assimilation using an open community hydrologic benchmark data set that is curated by the US National Center for Atmo-
spheric Research (NCAR). This Catchment Attributes and Meteorological Large Sample data set (CAMELS; Newman et al.,
2015; Addor et al., 2017) consists of daily meteorological and daily discharge data from 671 catchments in CONUS ranging
in size from 4 $km^2$ to 25,000 $km^2$ that have largely natural flows and long streamflow gauge records (1980-2014). Again, to
75 be consistent with previous studies (Kratzert et al., 2019c, 2021; Klotz et al., 2021; Gauch et al., 2021b; Newman et al., 2017;
Frame et al., 2021), we used the 531 of 671 CAMELS catchments that were chosen for model benchmarking by Newman
et al. (2017), who removed basins with (i) large discrepancies between different methods of calculating catchment area, and
(ii) areas larger than 2,000 $km^2$.

    CAMELS includes daily discharge data from the USGS Water Information System, which are used as training and evaluation
target data. CAMELS also includes several daily meteorological forcing data sets (Daymet, NLDAS, Maurer) that are used as
model inputs. Following Kratzert et al. (2021) we used all three data sets as inputs. CAMELS also includes several static
catchment attributes related to soils, climate, vegetation, topography, and geology (Addor et al., 2017) that we used as input
features – we used the same input features (meteorological forcings and static catchment attributes) that were listed in Table 1
by Kratzert et al. (2019c), and this table is repordiced in Appendix F.



## 2.2 Models

In total, we trained 61 LSTM models. One of these was a pure simulation model with no lagged streamflow data as inputs. This "simulation" model was used as a baseline for benchmarking performance of both DA and AR, and was also the base model used for data assimilation. 60 of the LSTMs were AR models: we tested 6 different lag times for ingesting near-real-time streamflow data (1, 2, 3, 5, 7, and 10 days), each associated with 10 different fractions of missing data (0% through 90%). Lagged streamflow data was withheld randomly throughout the input time series, and the strategy that we used for handling missing data is described in Sect. 2.2.2. We did not consider other types of missing data (i.e., meteorological forcings or basin attributes) because they are not central to the question at hand (how best to use lagged streamflow observations where those are available), and missing meteorological inputs and missing basin attributes are not common in operational models (most operational hydrology models require meteorological data at every timestep).

### 2.2.1 Training

Daily meteorological forcing data and static catchment attributes were used as input features for all models, and daily streamflow records were used as training targets with a normalized squared-error loss function that does not depend on basin-specific mean discharge (i.e., to ensure that large and/or wet basins are not over-weighted in the loss function):

$$\text{NSE*} = \frac{1}{B} \sum_{b=1}^{B} \sum_{n=1}^{N} \frac{(\widehat{y}_n - y_n)^2}{(s(b) + \epsilon)^2}. \tag{1}$$

$B$ is the number of basins, $N$ is the number of samples (days) per basin $B$, $\widehat{y}_n$ is the prediction for sample $n$ ($1 \leq n \leq N$), $y_n$ is the corresponding observation, and $s(b)$ is the standard deviation of the discharge in basin $b$ ($1 \leq b \leq B$), calculated from the training period (see, Kratzert et al., 2019c).

All models were trained using the training and test procedures outlined by Kratzert et al. (2019c). We trained for 30 epochs using sequence-to-one prediction to allow for randomized, small minibatches. We used a minibatch size of 256 and, due to sequence-to-one training, each minibatch contained (randomly selected) samples from multiple basins. We used 128 cell states and a 365-day sequence length. Input and target features were pre-normalized by removing bias and scaling by variance. Gradients were clipped to a global norm (per minibatch) of 1. Heteroscedastic noise was added to training targets (resampled at each minibatch) with standard deviation of 0.005 times the value of each target datum. We used an ADAM optimizer with a fixed learning rate schedule with initial learning rate of 1e-3 that decreased to 5e-4 after 10 epochs and 1e-4 after 25 epochs. Biases of the LSTM forget gate were initialized to 3 so that gradient signals persisted through the sequence from early epochs (Gers et al., 2000). All models were trained on data from all 531 CAMELS catchments simultaneously. The training period was October 1, 1999 through September 30, 2008 and the test period was October 1, 1989 through September 30, 1999.

### 2.2.2 Autoregression

The strategy that we used to deal with missing data in AR models was to replace missing lagged streamflow data with model-predicted streamflow data at the same lag time. This is related to a standard ML technique for training recursive models,





discussed in Appendix A. Autoregression models were thus trained with two inputs in addition to the CAMELS data inputs described in Sect. 2.1: (i) streamflow lagged by some number of days (different lag times) and (ii) a binary input flag that represents whether any particular autoregressive input came from observation or from previous model predictions. The binary flag allows the model to differentiate between observed vs. simulated autoregressive inputs.

### 2.2.3 Data Assimilation

The theory behind using backpropagation through tensor networks to perform variational data assimilation is given in Appendix B - this is essentially a direct implementation of standard variational DA using tensor networks.

Data assimilation was performed during the test period on the "simulation" LSTM outlined in Sect. 2.2. Unlike training an LSTM, where all training data from all catchments must be used together to train a single model (Nearing et al., 2020; Gauch et al., 2021b), data assimilation is independent between basins. As such, the loss function we used for data assimilation was a regularized Mean Squared Error (MSE) (for more details on the loss function used for data assimilation, see Appendix C)

We used the ADAM optimizer for data assimilation with a dynamic learning rate that started at 0.05 and decreased by a factor of 0.1 each time the update step loss failed to decrease. We used an assimilation window of 5 timesteps (updating the cell state at timesteps $t - 5$), 1000 update steps with early stopping criteria if the learning rate decreased below $1e - 6$, and we did not use any regularization. The search used to find these hyperparameter values for data assimilation is reported in Appendix E.

### 2.3 Testing & Evaluation

Following previous studies (cited in Sect. 2.1), we report a number of hydrologically relevant performance metrics, listed in Table 1. We report all of these metrics for the simulation model (no lagged streamflow data as inputs), and also for data assimilation and autoregression models. AR models were trained with fractions of missing data (withheld randomly) between 0% and 90% and tested on data with different fractions of missing data. This was done to understand what effect the training data fraction has on performance. After choosing an appropriate fraction of missing data for training AR models, these models were trained with streamflow inputs that had varying lag times (between 1 and 10 days). Both AR and DA models were tested with different fractions of (randomly) withheld lagged streamflow input data and different lag times, however all metrics were calculated on all streamflow observations within each basin during the entire test period, even when some of the lagged streamflow data were withheld as inputs.

## 3 Results

### 3.1 Training AR Models with Missing Data

Figure 1 compares median (over test periods in 531 basins) NSE values from AR models trained with 10 missing data fractions, each tested on 10 different missing data fractions. The primary signal in these results is that AR models lose accuracy as the



**Table 1.** Overview of evaluation metrics

| Metric | Description | Reference |
|---|---|---|
| NSE[i] | Nash-Sutcliff efficiency | Eq. 3 in Nash and Sutcliffe (1970) |
| KGE[ii] | Kling-Gupta efficiency | Eq. 9 in Gupta et al. (2009) |
| Pearson-r | Pearson correlation between observed and simulated flow | |
| $\alpha$-NSE[iii] | Ratio of standard deviations of observed and simulated flow | From Eq. 4 in Gupta et al. (2009) |
| $\beta$-NSE[iv] | Ratio of the means of observed and simulated flow | From Eq. 10 in Gupta et al. (2009) |
| Peak-Timing-Error[v] | Mean peak time lag (in days) between observed and simulated peaks | Appendix D |
| Peak-Timing-Abs-Error[vi] | Mean peak absolute time lag (in days) between observed and simulated peaks | Appendix D |
| Missed-Peaks[vii] | Number of observed peaks without simulated peaks within 1 day | Appendix D |
| Peak-Abs-Bias[viii] | Number of observed peaks without simulated peaks within 1 day | Appendix D |

[i]: *Nash-Sutcliffe efficiency:* $(-\infty, 1]$, *values closer to one are desirable.*

[ii]: *Kling-Gupta efficiency:* $(-\infty, 1]$, *values closer to one are desirable.*

[iii]: *$\alpha$-NSE decomposition:* $(0, \infty)$, *values close to one are desirable.*

[iv]: *$\beta$-NSE decomposition:* $(-\infty, \infty)$, *values close to zero are desirable.*

[v]: *Peak-Timing-Error:* $(-\infty, \infty)$, *values close to zero are desirable.*

[vi]: *Peak-Timing-Abs-Error:* $[0, \infty]$, *values close to zero are desirable.*

[vii]: *Missed-Peaks:* $[0, 1]$, *values close to zero are desirable.*

[viii]: *Peak-Abs-Bias:* $[0, \infty)$, *values close to zero are desirable.*

fraction of missing lagged streamflow data in the test period increases. In general, training with fewer missing data is better if the fraction of missing data in the test period is also low. However, if the fraction of missing data in the test period is high, then it is better to train with more missing data.

For the remainder of our experiments, we chose to benchmark AR models trained with 50% missing lagged streamflow
inputs. This represents a compromise between training with too many or too few missing data that only degrades below the (median) accuracy of the pure simulation model with a missing data fraction of 90%.

## 3.2 Benchmarking

Table 2 lists the performance metrics for all models with a lag of one day and no missing data. The major takeaways from these statistics are that both variational assimilation and autoregression improved over the base LSTM model but autoregression
was better. Autoregression improved the median NSE (across test periods in 531 basins) by $\sim$10%, whereas data assimilation improved the median NSE by $\sim$8%. In general, autoregression with no missing data performed better than data assimilation across all metrics, and also across most basins (Fig. 2). Note that autoregression trained with and without any missing lagged streamflow data performed similarly (Fig. 2).

As a point of comparison with previous work, (Feng et al., 2020) reported that autoregression improved LSTM median NSE
by $\sim$19% (from 0.714 to 0.852), whereas we saw improvement to median NSE of $\sim$10%. The primary difference between that





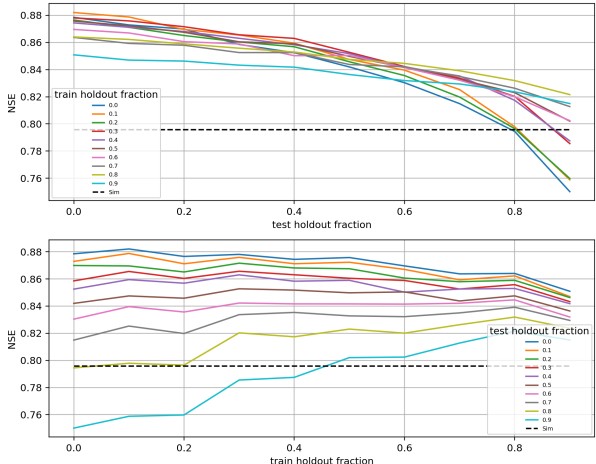

**Figure 1.** Median NSE scores of AR models trained and tested with different fractions of lagged streamflow data withheld. The two subplots show the same results, but organized by the amount of lagged streamflow data withheld during training vs. during testing.

**Table 2.** Results for 1-day lag and no missing data

| Metric | Simulation | AR w/o holdout | AR w/ holdout | Assimilation |
|---|---|---|---|---|
| NSE | 0.796 | 0.879 | 0.876 | 0.860 |
| KGE | 0.791 | 0.895 | 0.896 | 0.841 |
| Alpha-NSE | 0.875 | 0.937 | 0.953 | 0.875 |
| Pearson-r | 0.902 | 0.940 | 0.939 | 0.933 |
| Beta-NSE | -0.030 | -0.012 | 0.004 | -0.026 |
| Peak-Timing-Error | 0.000 | -0.115 | -0.129 | -0.206 |
| Peak-Timing-Abs-Error | 0.206 | 0.304 | 0.300 | 0.360 |
| Missed-Peaks | 0.342 | 0.238 | 0.265 | 0.250 |
| Peak-Abs-Bias | 0.285 | 0.212 | 0.221 | 0.222 |

previous study and ours is that our baseline LSTMs were better (median NSE of 0.800 vs. 0.714) due to the fact that the models reported in this study used multiple forcing data products (Kratzert et al., 2021).

Feng et al. (2020) also reported that autoregression was less informative in flashy basins, and there is a similar effect in our results (Appendix F reports similar efforts here to correlate differences between autoregression and data assimilation with basin attributes related to climate, geology, soils, vegetation). While both autoregression and assimilation improved the average absolute peak timing error and also allowed the model to miss fewer peak-flow events, the peak timing error with both





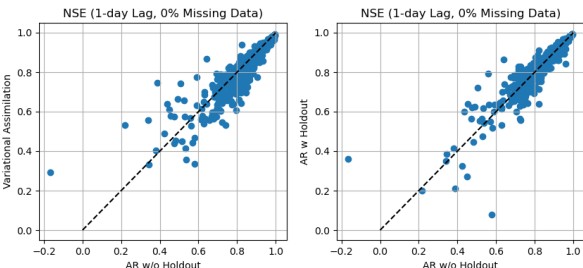

**Figure 2.** Comparison of per-basin NSEs with an observation lag of one day and no missing streamflow input data: (left) autoregression trained with no holdout data vs. variational assimilation and (right) autoregression with vs. without 50% of lagged streamflow data withheld during training.

autoregression and data assimilation was always negative due to the fact that the model receives delayed information from lagged streamflow about any peak event that it might otherwise miss predicting from meteorological data alone.

Fig. 3 compares the median NSE scores (over 531 basins) between the four models as a function of lag time in days and fraction of missing lagged streamflow data in the test period. Autoregression is always better than data assimilation, but if the autoregression model is not trained with a fraction of missing data (here 50%), then performance decreases if there is missing data in the test period. In this case, the autoregression model can perform worse than a simulation model with no lagged streamflow data. If the autoregression model is trained with an input flag to indicate whether a particular lagged streamflow value is from observation or simulation, then autoregression almost always improves on the baseline simulation model and is almost always better than data assimilation (up to large values of missing data). Similar figures for all metrics in Table 1 are given in Appendix G.

## 4 Conclusions & Discussion

Data assimilation is necessary in order to use certain types of data to "drive" dynamical systems models. For example, if a model is based on some conceptual understanding of a physical system (like a conceptual process-based rainfall-runoff model), then the only way to use observations of system states and outputs is via an inverse method. DA is a class of inverse methods that project information onto the states of a dynamical systems model. DA is often complicated to set up (e.g., choosing parameters to represent uncertainty distributions, sampling procedures, etc.), and often requires simplifying assumptions that cause significant information loss (e.g., Nearing et al., 2018). ML models do not suffer from these same limitations – it is possible to simply train the models to use whatever data is available as inputs. This has several advantages over DA, including ease of use and computational efficiency. The forward (ML) approach also allows the model to learn the best way to use the new input data streams directly, instead of through inverse methods. We've seen hints in previous work that the latter allows for more information extraction from data (Nearing et al., 2013).





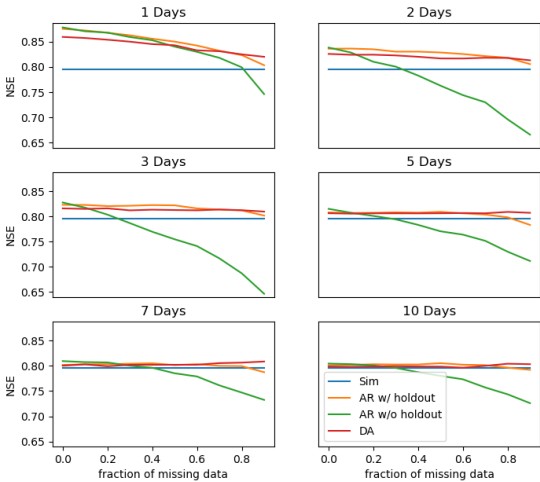

**Figure 3.** Median NSE over 531 basins of four models (simulation, autoregression trained with and without holdout data, and data assimilation) as a function of lag time in days and fraction of missing lagged streamflow data in the test period.

To reiterate from the introduction, it is impossible to test every AR or every DA method in a single study. We also do not know of any benchmarking study in the hydrology literature that directly compares different DA methods (typically a single
method is applied and tested). Because of this, we cannot make conclusive statements about whether all DA methods might under-perform compared to AR in a ML context, however we have no reason to suspect that other DA methods might perform better than variational DA on large sample data sets. Given that DA systems generally require large investments of time and resources to develop and tune, the purpose of this current study is to give a sense of whether this type of investment might be worthwhile given that there is another option (AR) for ML-based models. The AR method we tested was exceptionally simple
to implement and significantly out-performed the more complicated DA method. As such, the primary actionable conclusion of this study is that we do *not* recommend DA as a strategy for using near-real-time streamflow data. We would instead recommend investing effort into better strategies for AR (e.g., different methods for dealing with missing data, using multiple data streams at potentially different spatiotemporal resolutions, etc.).

*Code and data availability.* All code used for this study is available at https://github.com/grey-nearing/lstm-data-assimilation.
CAMELS data is available at https://ral.ucar.edu/solutions/products/camels, with extensions to 2014 at
https://www.hydroshare.org/resource/0a68bfd7ddf642a8be9041d60f40868c/ and
https://www.hydroshare.org/resource/17c896843cf940339c3c3496d0c1c077/.





*Author contributions.* Daniel Klotz had the original idea for backpropagation-based data assimilation. Grey Nearing formulated this as a form of variational data assimilation, wrote the data assimilation code, performed hyperparameter tuning, and conducted all experiments except correlations with basin attributes, which were done by Jonathan Frame. Martin Gauch contributed to the design of experiments (suggested to implement the input data flag for autoregression). Martin Gauch and Frederik Kratzert integrated the data assimilation code into the NeuralHydrology codebasde. Grey Nearing wrote the paper.

*Competing interests.* The authors declare no competing interests

*Acknowledgements.* Frederik Kratzert was supported by a Google Faculty Research Award (PI: Sepp Hochreiter). Martin Gauch was supported by the Linz Institute of Technology DeepFlood project. Daniel Klotz was supported by Verbund AG.

## Appendix A:  Related ML Techniques for Autoregressive Modeling

The strategy we use for handling missing data in AR models is loosely related to a class of techniques used commonly to train recursive neural networks called *teacher-forcing* methods (Williams and Zipser, 1989). Teacher-forcing methods substitute model outputs from timestep $t$ as inputs into the network at timestep $t + 1$ with observations during training (not during inference). This was originally used to avoid backpropagating through time, and the method is sensitive to differences between train and test samples which can result in divergent behavior when the model is run recursively during inference.

Our strategy for dealing with missing data in AR models does not replace the cell state (the recursive state of the LSTM)during training, but does run the risk of over-training to observed lagged streamflow inputs in cases where these data are sparse during inference (Figure 1). Our solution to this problem is to train with a combination of observed and simulated lagged streamflow inputs, which is a form of *scheduled sampling* (Bengio et al., 2015). This seems to work in this case (see Sect. 3.1). Another class of approaches to solving this problem are *professor-forcing* methods (Lamb et al., 2016), which uses adversarial learning to encourage a teacher-forcing network (i.e., trained with only observed inputs) to match the fully recursive model. This could be applied to the streamflow problem if AR models were to exhibit divergent behavior that cannot be solved with scheduled sampling.

## Appendix B:  Variational Data Assimilation in LSTMs

The method of data assimilation that we will use in this paper is a type of variational assimilation. Variational assimilation works as follows (Rabier and Liu, 2003). We begin with a model that has time-dependent states, $\boldsymbol{c}[t]$ that determine a time-dependent observable, $\boldsymbol{y}[t]$, through a (potentially nonlinear) observation operator $\mathcal{H}$, up to random error $\boldsymbol{\varepsilon_y}[t]$:

$$\boldsymbol{y}[t] = \mathcal{H}(\boldsymbol{c}[t]) + \varepsilon_y[t] \tag{B1}$$





The model itself propagates through time according to some state transition function, $\mathcal{M}$, that operates on the state at the previous timestep and model inputs, $\boldsymbol{x}[t]$:

$$\boldsymbol{c_b}[t] = \mathcal{M}(\boldsymbol{c}[t-1], \boldsymbol{x}[t]) \tag{B2}$$

The *background state*, $\boldsymbol{c_b}[t]$, is the state estimated by model $\mathcal{M}$ at time $t$ without performing assimilation at time $t$ (although assimilation may have been performed at previous timesteps). The true (but unknown) state of the system is assumed to be

equal to the background state up to random error $\varepsilon_c$:

$$\boldsymbol{c}[t] = \boldsymbol{c_b}[t] + \boldsymbol{\varepsilon_c}[t] \tag{B3}$$

Notating observations and states as drawn from distributions $p_y$ and $p_c$, we condition the model state on observations at time $t$ as:

$$p(\boldsymbol{c}[t]|\boldsymbol{y}[t], \boldsymbol{c_b}[t]) \propto p_y(\boldsymbol{y}[t]|\boldsymbol{c}[t])p_c(\boldsymbol{c}[t]|\boldsymbol{c_b}[t]). \tag{B4}$$

The maximum likelihood estimate of the state vector is found by minimizing the negative log likelihood associated with Eq. B4. For example, if the state and observation errors ($\boldsymbol{\varepsilon_c}$ and $\boldsymbol{\varepsilon_y}$) are assumed to be normally distributed, the resulting loss function is:

$$\mathcal{J}(\boldsymbol{c}[t]) = (\boldsymbol{c}[t] - \boldsymbol{c_b}[t])^T \boldsymbol{B}^{-1}(\boldsymbol{c}[t] - \boldsymbol{c_b}[t]) + (\boldsymbol{y}[t] - \mathcal{H}(\boldsymbol{c}[t]))^T \boldsymbol{R}^{-1}(\boldsymbol{y}[t] - \mathcal{H}(\boldsymbol{c}[t])) \tag{B5}$$

where $\boldsymbol{B}$ and $\boldsymbol{R}$ are covariances of the state and observation errors, respectively. Analytical solutions are known for the special

case when $\mathcal{H}(\cdot)$ is linear. Eq. B5 can be understood as a regularized loss function acting on target variables that is to be maximized with respect to the model states. If any component of this is not linear and Gaussian, then $\mathcal{J}(\cdot)$ must be minimized numerically.

The LSTM is described by the following equations:

$$\boldsymbol{i}[t] = \sigma(\boldsymbol{W_i}\boldsymbol{x}[t] + \boldsymbol{U_i}\boldsymbol{h}[t-1] + \boldsymbol{b_i}) \tag{B6}$$

$$\boldsymbol{f}[t] = \sigma(\boldsymbol{W_f}\boldsymbol{x}[t] + \boldsymbol{U_f}\boldsymbol{h}[t-1] + \boldsymbol{b_f}) \tag{B7}$$

$$\boldsymbol{g}[t] = \tanh(\boldsymbol{W_g}\boldsymbol{x}[t] + \boldsymbol{U_g}\boldsymbol{h}[t-1] + \boldsymbol{b_g}) \tag{B8}$$

$$\boldsymbol{o}[t] = \sigma(\boldsymbol{W_o}\boldsymbol{x}[t] + \boldsymbol{U_o}\boldsymbol{h}[t-1] + \boldsymbol{b_o}) \tag{B9}$$

$$\boldsymbol{c}[t] = \boldsymbol{f}[t] \odot \boldsymbol{c}[t-1] + \boldsymbol{i}[t] \odot \boldsymbol{g}[t] \tag{B10}$$

$$\boldsymbol{h}[t] = \boldsymbol{o}[t] \odot \tanh(\boldsymbol{c}[t]), \tag{B11}$$

$\boldsymbol{x}[t]$ are again the model inputs at time $t$, and $\boldsymbol{i}[t]$, $\boldsymbol{f}[t]$ and $\boldsymbol{o}[t]$ refer to the *input gate*, *forget gate*, and *output gate* of the LSTM, respectively. $\boldsymbol{g}[t]$ are the *cell inputs*, $\boldsymbol{h}[t-1]$ are the LSTM outputs, which are also called the *recurrent input* because these are used as inputs to all gates in the next timestep. $\boldsymbol{c}[t-1]$ are the *cell state* from the previous time step. Similar to dynamical systems models, the cell state, $\boldsymbol{c}[t]$ tracks the time-evolution of the system.





Model-predicted streamflow values comes from a *head layer*, and many LSTM studies in hydrology (e.g., Kratzert et al.,
2018) have used a linear (dense) head layer:

$$y[t] = \boldsymbol{h}[t]\boldsymbol{w_h} + b_h \qquad (B12)$$

Eqs. B11 and B12 effectively define the observation operator for data assimilation (Eq. B1). Notice that observations $\boldsymbol{y}[t]$ are
not linear functions of the cell state (through the hyperbolic tangent operator in Equation B11), so no analytical solution to
maximizing eq B4 or minimizing Eq. B5 exists. We therefore must minimize $\mathcal{J}(\boldsymbol{c}[t])$ numerically, which requires gradients
with respect to the cell states.

In any deep learning model, the various weights, $\boldsymbol{W_*}$, and biases, $\boldsymbol{b_*}$, are trained by minimizing a training loss function $L(\cdot)$
using backpropagation along gradient chains like:

$$\frac{\delta L(\boldsymbol{x}[0:t], \boldsymbol{y}[0:t])}{\delta w_{*,j}} = \frac{\delta L(\boldsymbol{x}[0:t], \boldsymbol{y}[0:t])}{\delta h_k[t]} \times \frac{\delta h_k[t]}{\delta c_l[t]} \times \frac{\delta c_l[t]}{\delta *} \times ... \times \frac{\delta *}{\delta w_{*,j}}. \qquad (B13)$$

where the subscripts $j$ and $k$ (e.g., $w_{*,j}$, $h_k[t]$) indicate an arbitrary components of vectors or matrices (e.g., $\boldsymbol{W_*}$ like $\boldsymbol{W_i}$,
$\boldsymbol{W_f}$, $\boldsymbol{W_o}$, or $\boldsymbol{W_g}$). $\boldsymbol{h}[t-1]$ are again the LSTM outputs (recurrent inputs), and the ellipses indicate that the network may have
arbitrary depth. Eq. B13 is a simple derivative chain rule that any machine learning software library calculates automatically
through the entirety of whatever tensor network defines a particular model. Almost all deep learning models are trained by
backpropagating information through this type of gradient chain. Every time that the training loss function $L(\cdot)$ is calculated
on a series of model inputs, $\boldsymbol{x}[0:t]$, and outputs, $\boldsymbol{y}[0:t]$, the values of all weights and biases in the model are updated based
on perturbing in a direction that will decrease the loss according to these partial derivatives.

Notice that gradient chains like Eq. B13 necessarily include partial derivatives of the loss function with respect to features
in the model that are not weights and biases. As an example, the partial derivative of loss $L$ with respect to weights in the
input gate, $\boldsymbol{W_i}$, requires derivatives with respect to the cell states, $\boldsymbol{c}[t]$. To perform data assimilation, we can simply break the
gradient chains to get partial derivatives of a loss function with respect to cell states like:

$$\frac{\delta L(\boldsymbol{x}[0:t], \boldsymbol{y}[0:t])}{\delta w_{i,j}} = \frac{\delta L(\boldsymbol{x}[0:t], \boldsymbol{y}[0:t])}{\delta h_k[t]} \times \frac{\delta h_k[t]}{\delta c_l[t]} \qquad (B14)$$

Gradient chains like Equation B13 are used when training deep learning models. In this case, the loss function is calculated
over a large number of historical data points (sometimes using minibatches). We want to be able to use streamflow observation
data as it becomes available in near-real-time, which means that we want to use gradient chains like Equation B14 during
inference rather than during training. These take the following form:

$$\frac{\delta \mathcal{L}(x[0:t], y[t-s:t])}{\delta c_l[t]} = \frac{\delta \mathcal{L}(x[0:t], y[0:t])}{\delta h_k[t]} \times \frac{\delta h_k[t]}{\delta c_l[t]} \times \frac{\delta c_l[t]}{\delta *} \times ... \times \frac{\delta *}{\delta c_l[t-s]}. \qquad (B15)$$

The primary difference between Equations B14 and B15 is that the loss function is calculated over observations a finite time
period, $s$, into the past, $\delta L(x[0:t], y[t-s:t])$, and used to update cell states at the start of the current observation period. After
the model is fully trained, and while it is running in forward mode to make new predictions, we can at any point calculate a
loss function like $\delta \mathcal{L}(x[0:t], y[t-s:t])$ and use this to update the cell states of the LSTM using gradient chains like B15.





## Appendix C:  Data Assimilation Loss Function

The loss function used for assimilation (i.e., $\mathcal{L}(\cdot)$ in Eqs. B15 or **??**) do not need to be the same loss functions used for training (i.e., $L$ in Eq. B13). The derivatives that result from gradient chains in a deep learning tensor network can be calculated with respect to any loss function. Additionally, any loss function can be augmented with regularization – for example, to ensure that the updated cell states do not deviate too much from the values that are estimated by the trained model. The $R$ and $B$ matrices in Eq. B5 are an example of this type of regularization, and it is trivial to use this (or any other) type or regularization in the data assimilation loss function.

$$\mathcal{J}(\boldsymbol{c}[t]) = \alpha_c(\boldsymbol{c}[t] - \boldsymbol{c_b}[t])^T(\boldsymbol{c}[t] - \boldsymbol{c_b}[t]) + \alpha_y(\boldsymbol{y}[t:t+s] - \widehat{\boldsymbol{y}}[t:t+s])^T(\boldsymbol{y}[t:t+s] - \widehat{\boldsymbol{y}}[t:t+s]) \tag{C1}$$

Coefficients $\alpha_c$ and $\alpha_y$ are constants that are analogous to the $B$ and $R$ terms in Eq. B5, and $s$ is an integer number of timesteps that defines an assimilation window so that the cell state at time $t$ is updated based on observations through time $t + s$. Gradient chains like Eq. B15 do not look forward in time in the sense that the derivative of $\mathcal{L}$ with respect to $\boldsymbol{c}[t]$ does not depend on any observation prior to time $t$. This means that the assimilation loss Eq. C1 is general in $s$.

## Appendix D:  Description of Peak-Timing Metrics

Peak timing metrics were calculated by first locating all peaks in the observation and simulation time series that satisfy the following two criteria: (1) observed and simulated peaks must be at least 30 days apart, and (2) peaks must be above the $95^{th}$ flow percentile in a given basin. We report four statistics:

1. The fraction of observed peaks that the model predicts (above the $95^{th}$ flow percentile) within 1 day of an observed peak.

2. The average timing error (in unit days) calculated only on peaks that the model predicts.

3. The average absolute timing error (in unit days) calculated only on peaks that the model predicts.

4. The absolute percent bias of peak flow calculated only on peaks that the model predicts.

## Appendix E:  Hyperparameter Tuning

Hyperparameter tuning for data assimilation was done with a validation period (1980-1989) that is distinct from both the training (1999-2008) and test periods (1989-1999) outlined in Sect. **??**. Due to computational expense, hyperparameter tuning was done on a subset of 20 basins out of the 531 used for the rest of the study. The grid search is outlined in Table E1. This grid search resulted in the final data assimilation hyperparameters listed in the right-most column of Table E1.



**Table E1.** Data assimilation hyperparameter tuning grid search and final values.

| Hyperparameter | Grid Search | Best Value |
|---|---|---|
| Initial Learning Rate | [0.05, 0.01] | 0.05 |
| Learning Rate Epochs Drop | [1, 5, 10, 50, inf] | inf |
| Learning Rate Drop Factor | [0.9, 0.5, 0.1] | 0.1 |
| Assimilation Window[i] | [1, 5, 20] | 5 |
| Assimilation History | [1, 5, 20, 50] | 20 |
| Regularization[ii] | [0, 0.01, 0.1, 1, 2] | 0 |

[i]: *This is $s$ from Eq. C1*

[ii]: *This is $\alpha_c$ from Eq. C1*





We tested a learning rate scheduler that dropped the learning rate every $N$ epochs. This is the *Learning Rate Epochs Drop* hyperparameter in Table E1 and did not improve performance. Because we used sequence-to-one prediction (Sect. 2.2.1), we did not perform assimilation through the entire time series. The *Assimilation Window* hyperparameter determines how far back in each sequence (before time of prediction) we start assimilation.

In our setup $\alpha_c$ (see Equation C1) was set as a hyperparameter and not trained directly, although learning this parameter through backpropagation is possible. Our hyperparameter search returned a value of $\alpha_c = 0$ (and and implied value of $\alpha_y = 1$), meaning that regularizing the loss function was not helpful.

## Appendix F: Performance by Basin Attributes

We tested whether it was possible to predict where DA or AR might offer the most benefit by using CAMELS catchemnt
attributes (Addor et al., 2017). These attributes and their abbreviations are listed in Table F1. We used random forest models trained with static catchment attributes as inputs using k-fold cross validation to measure the predictability of the increase or decrease in test-period NSE scores at individual basins. The objective was to determine which types of hydrological characteristics determine the value or information content of lagged streamflow data.

    Results for this analysis are given in Figure F1. The top subplots of Figure F1 illustrate the ability to predict test-period NSE
scores from basin attributes in the three models (simulation, AR, DA), and the bottom subplots illustrate the ability to predict differences between test-period NSE scores from the different models. Kratzert et al. (2019c) found that forest fraction was strongly correlated with a predictor of the basin similarity mapping in an LSTM, and we see a similar effect in the top left subplot of Figure F1 (forest fraction is the second highest predictor of the skill of the simulation model).

    Feng et al. (2020) reported that AR was less informative in flashy basins, and we see some evidence of that effect here: in
particular, the frequency of low precipitation days was the strongest predictor of AR skill such that lower fractions of rainfall occurring in low intensity events corresponds with higher AR skill. Similarly, basin area was the third strongest predictor of AR skill, with AR being better in larger basins.

    Snow fraction was the second strongest predictor of skill for both AR and DA, whereas this was not a strong predictor of skill in the pure simulation model. Kratzert et al. (2019a) showed that LSTMs can learn to store and release snow (without
seeing snow data), however snowpack introduces correlations in streamflow time series that are exploited by both DA and AR.

    The basin attributes that were the most important for determining NSE *improvements* due to both DA and AR (bottom center and bottom right subplots of Figure F1) were (i) mean annual precipitation and (ii) high precipitation frequency. High precipitation frequency was positively correlated with both AR and DA performance improvements. The reason for this is error in the rainfall data – AR and DA allow the model to effectively "see" streamflow events that occur due to unobserved or under-
observed rainfall, although it takes one timestep for the model to register that a large event happened. This helps the model to avoid large errors for events with long recession curves. In general, average precipitation was negatively correlated with improvements due to incorporating lagged streamflow data, since precipitation events in general reduce autocorrelation in the streamflow time series. This effect appears worse for DA than AR (bottom left subplot), although this is a weak signal because





**Table F1.** Table of static catchment attributes. Descriptions taken from Addor et al. (2017)

| | |
|---|---|
| p_mean | Mean daily precipitation. |
| pet_mean | Mean daily potential evapotranspiration. |
| aridity | Ratio of mean PET to mean precipitation. |
| p_seasonality | Seasonality and timing of precipitation. Estimated by representing annual precipitation and temperature as sin waves. Positive (negative) values indicate precipitation peaks during the summer (winter). Values of approx. 0 indicate uniform precipitation throughout the year. |
| frac_snow_daily | Fraction of precipitation falling on days with temperatures below $0°C$. |
| high_prec_freq | Frequency of high precipitation days (>= 5 times mean daily precipitation). |
| high_prec_dur | Average duration of high precipitation events (number of consecutive days with >= 5 times mean daily precipitation). |
| low_prec_freq | Frequency of dry days (< 1 mm/day). |
| low_prec_dur | Average duration of dry periods (number of consecutive days with precipitation < 1 mm/day). |
| elev_mean | Catchment mean elevation. |
| slope_mean | Catchment mean slope. |
| area_gages2 | Catchment area. |
| forest_frac | Forest fraction. |
| lai_max | Maximum monthly mean of leaf area index. |
| lai_diff | Difference between the max. and min. mean of the leaf area index. |
| gvf_max | Maximum monthly mean of green vegetation fraction. |
| gvf_diff | Difference between the maximum and minimum monthly mean of the green vegetation fraction. |
| soil_depth_pelletier | Depth to bedrock (maximum 50m). |
| soil_depth_statsgo | Soil depth (maximum 1.5m). |
| soil_porosity | Volumetric porosity. |
| soil_conductivity | Saturated hydraulic conductivity. |
| max_water_content | Maximum water content of the soil. |
| sand_frac | Fraction of sand in the soil. |
| silt_frac | Fraction of silt in the soil. |
| clay_frac | Fraction of clay in the soil. |
| carb_rocks_frac | Fraction of the catchment area characterized as "Carbonate Sedimentary Rocks". |
| geol_permeability | Surface permeability (log10). |

we were generally unable to predict differences between the NSE scores of DA and AR ($r^2 = 0.12$; bottom right subplot in

Figure F1).





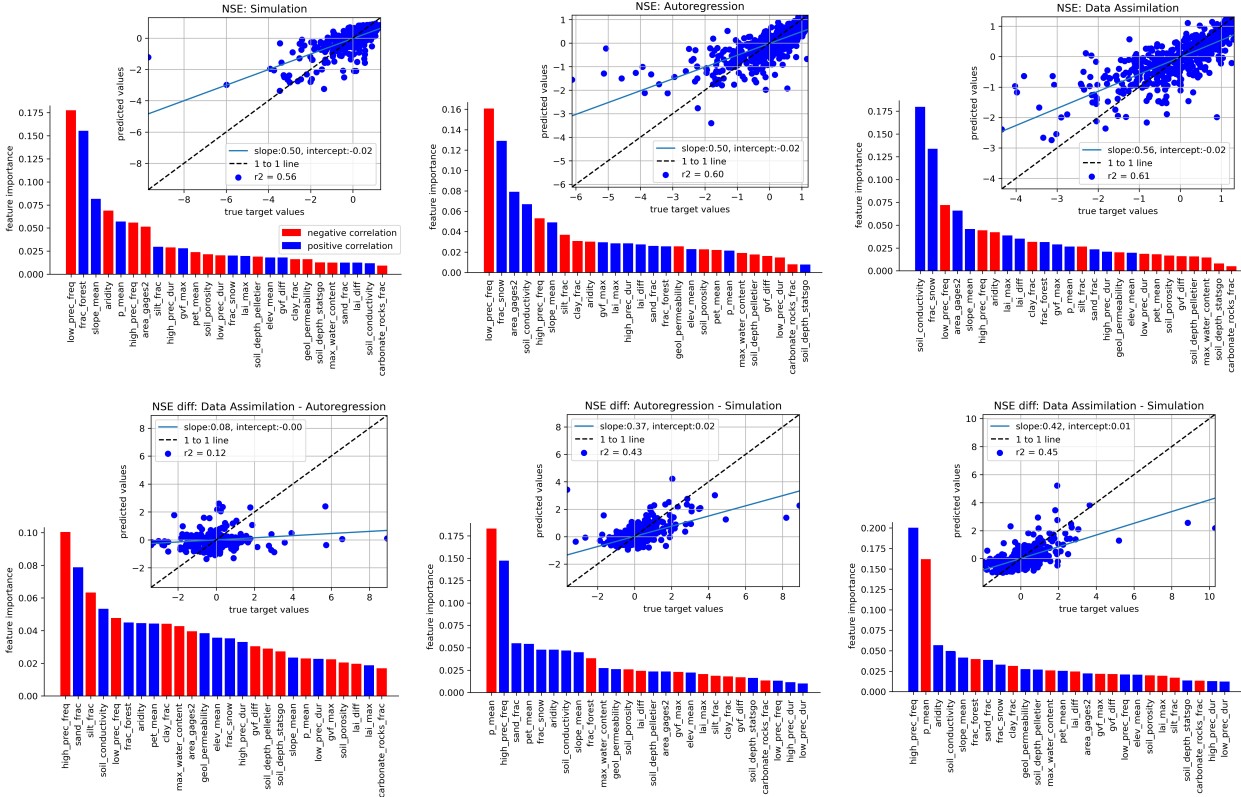

**Figure F1.** Scatterplots, $r^2$ metrics, and feature importances for predicting test-period NSE scores using different models (top subplots), as well as for predicting differences between models (bottom subplots). Bar charts show the feature importance for these predictions with blue indicating positive correlations between a given basin attribute and the NSE (or delta-NSE) and red indicating a negative correlation.

Figure F2 shows the spatial distribution of DA and AR NSE improvements relative to simulation. In both cases – but especially for DA – there is a group of basins in the Midwest and Southeast United States where performance was harmed by adding lagged streamflow data. We are unsure of the reason for this, but it warrants further exploration.

## Appendix G: All Metrics Figures

This appendix contains figures similar to Fig. 3 for all metrics listed in Table 1. These figures compare the median (over 531 basins) performance of four models (simulation, autoregression trained with and without holdout data, and data assimilation) as a function of lag time in days and fraction of missing lagged streamflow data in the test period.





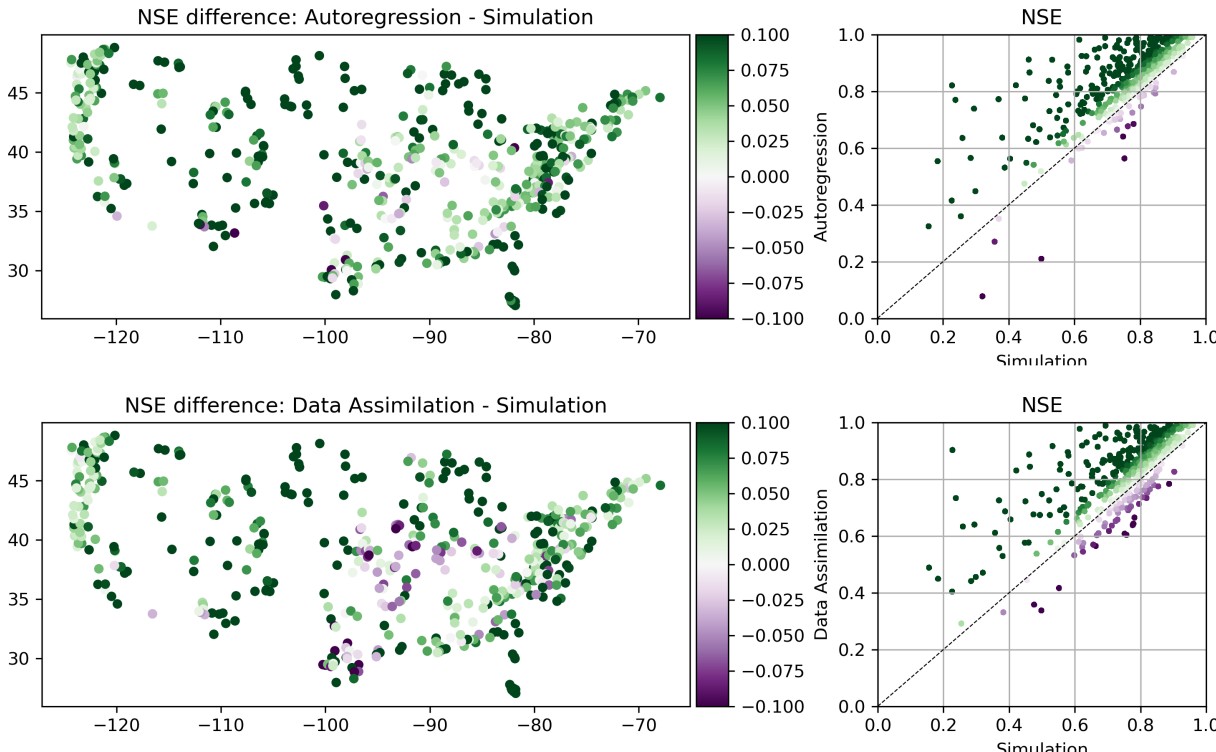

**Figure F2.** The performance difference (NSE score) between top: autoregression and the baseline simulation, and bottom: data assimilation and the baseline simulation.

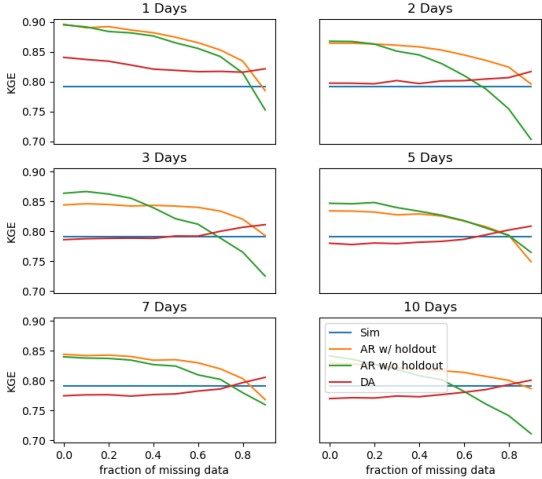

**Figure G1.** Same as Fig. 3 but for Kling-Gupta Efficiency.



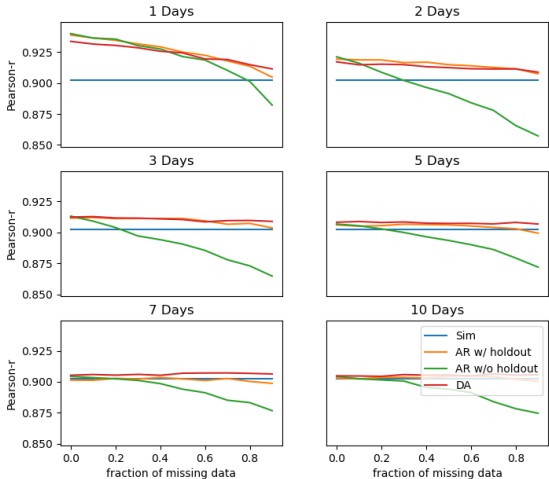

**Figure G2.** Same as Fig. 3 but for the Pearson correlation coefficient.

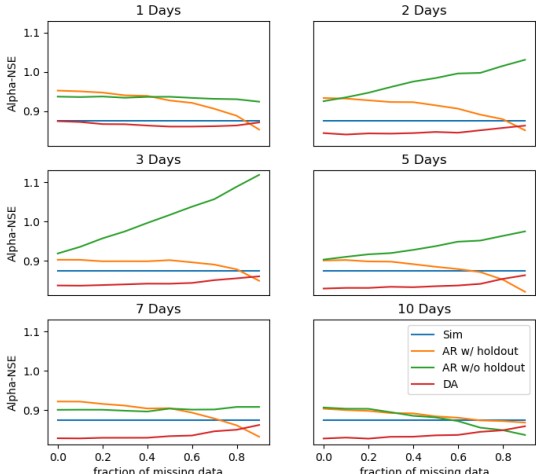

**Figure G3.** Same as Fig. 3 but for $\alpha - NSE$, which is the ratio of the standard deviation of the observed vs modeled hydrographs.





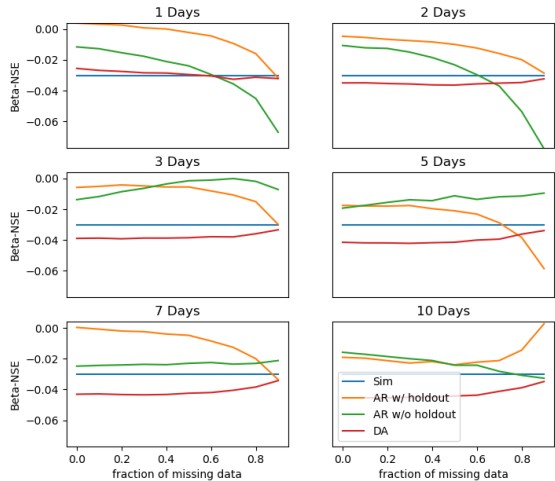

**Figure G4.** Same as Fig. 3 but for $\beta - NSE$, which is the ratio of the means of the observed vs modeled hydrographs.

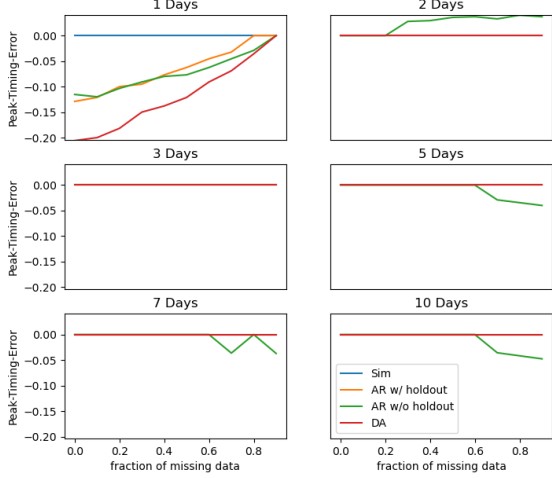

**Figure G5.** Same as Fig. 3 but for peak timing error (Appendix D).





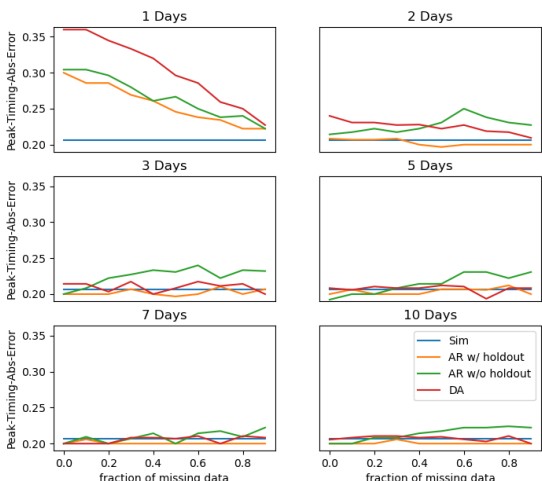

**Figure G6.** Same as Fig. 3 but for absolute peak timing error (Appendix D).

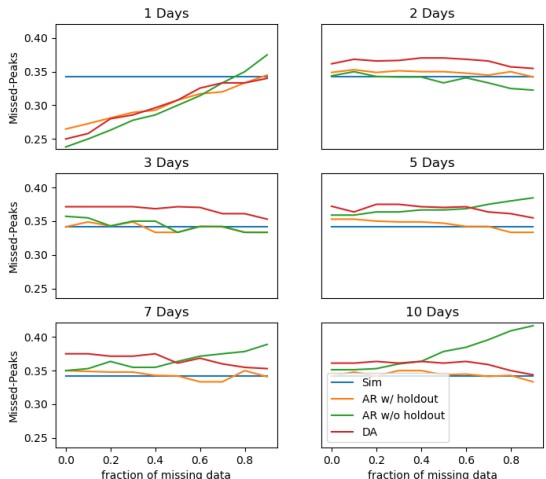

**Figure G7.** Same as Fig. 3 but for the fraction of missed peak flow events (Appendix D).





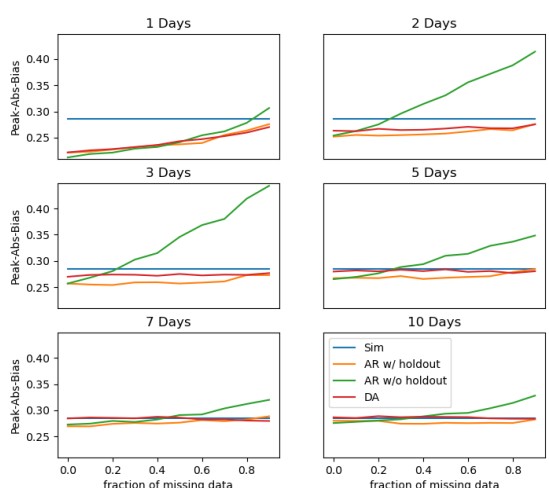

**Figure G8.** Same as Fig. 3 but for percent absolute bias (Appendix D).





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
