# Peer review of "Technical Note: Data assimilation and autoregression for using near-real-time streamflow observations in long short-term memory networks"

_Hydrology and Earth System Sciences, 2021_

## Author Comment (AC1)

**Reviewer #1**

Summary and Recommendation

Nearing et al. test how near-real time streamflow observations can effectively be used in Long Short-Term Memory (LSTM) rainfall runoff models. They compare an autoregression (AR) approach with a data assimilation (DA) approach and test, additionally, how sensitive AR is to random gaps in the data. The manuscript (MS) is easy to follow, well-structured and suits the scope of HESS. I particularly liked the comprehensive appendix in combination with a short MS. I think that this MS can be published after some minor revisions and provide only some smaller comments and questions below.

Sincerely

Ralf Loritz

Questions and comments:

- Reading the MS I would have like to see a couple of detailed results from three or four catchments where the AR or DA worked particular good or bad and what the (hydrological and ML) reason for this might be (to underpin the discussion of Appendix F and G). For instance, what could be the reason that DA and AR reduces the predictive performance of a few of your models (Fig. F2)? You state that: "*We are unsure of the reason for this, but it warrants further exploration.*" (Line 352) maybe zooming into one of the catchments could help to give a better explanation.

Thank you for the suggestion. We do agree with the notion of this comment. Indeed, We tried this extensively before submitting the manuscript and did not find a cohesive story to tell. We do not have a strong issue with this either way, and could be convinced to add this type of analysis, however we prefer not to because we didn't find any value in doing so. It's just extra text for very little payoff. Since this was a negative result, we only included the statistical analysis in appendix F. Again, happy to take this advice if the reviewer or editor thinks it's critical, however we did make this decision consciously, and we prefer not to.

- I find it a bit unrealistic how you added the missing data. I would assume that a broken gauging station is not working for a couple of days or maybe weeks in a row and wonder how this would alter your results (e.g. all streamflow data available for training but then two weeks or more only simulated data during testing with a closer focus on particular that period and not the entire testing period).

This is a good point. We will include this type of analysis in the revision (withhold time-continuous periods of data). Also, we will include an "ungauged basin" analysis, where we test both methods (DA, AR) on catchments where there is no gauge data at all during training and/or inference (we will use gauged catchments and withhold the data, so as to have data for evaluation).

- ● Showing how the variance or the Shannon entropy changes of your simulations in addition to the median would be interesting (Fig.1 and Fig.3). If it remains constant, I would mention that the spread of the predictions is not affected by the data availability.

Yes, this is a strange oversight on our part. We will include CDF plots in the revised manuscript.

Personal comment: Three of the seven Co-Authors have presumingly not contributed to this "technical note" as they are not mentioned in the author contribution section.

Thank you, we will add contributions by all authors in the revision.

---

## Author Comment (AC2)

**Reviewer #2**

The technical note compares two different techniques for using near-real-time streamflow observations to improve operational streamflow forecasts from LSTM rainfall-runoff models. The first technique ("autoregression", AR) adds lagged streamflow observations as predictor in the model. The second technique uses variational data assimilation (DA) to update model states within an assimilation window. The two techniques are compared on the CAMELS dataset, including experiments that artificially remove data to simulate scenarios with missing streamflow data.

The paper is generally well written, concise and to the point. The comparison between AR and DA is an interesting and novel contribution to the literature.

Comments:

1. The main conclusion is that "AR significantly out-performed the more complicated DA method" (line 195) and the authors therefore recommend against using DA (line 196). However, I feel the authors are overstating the results: differences in improved performance between AR (10%) and DA (8%) are relatively small, as also seen in Fig. 3 where the DA lines (red) and the AR lines (orange) are close.

We will add a significance test to the statement that the reviewer quoted. The central issue is that DA is more complicated than AR (due to the need for hypertuning both for training and inference, instead of just for training). DA would only be preferable if it performed significantly better, which it does not.

To be clear, there are other use cases for DA in LSTMs or other types of DL models (e.g., we are using it currently for a project that does image prediction), but we cannot see a use case for streamflow modeling unless someone can solve the problem that we were not able to solve … predicting in advance what type(s) of basins it might work well from available data.

2. On line 51 it is stated that "the purpose of this paper is to provide insight into trade-offs between DA and AR". I feel the paper doesn't entirely deliver on this. Yes, the two techniques are compared across a large number of basins, but the reader doesn't get a clear sense when to use which technique. Appendix F contains a regression analysis in this direction but concludes that "we were generally unable to predict differences between the NSE scores of DA and AR". Closer inspection by a human however may lead to some insights. E.g. it could be interesting to look in more detail at extreme cases: ones where AR significantly beats DA, and vice versa. For example figure 2 shows dots in the south that are green (good) for AR and purple (bad) for DA, and vice versa.

We did look at this, and did not find a story to tell. Explaining that we did not find a story to tell was the point of our statistical analysis in Appendix F. Our opinion was that including anecdotal examples didn't advance that story or improve the paper. We are happy to be persuaded, however we did consider this (and even included such examples in an early draft). But our opinion was that it just became clutter.

3. Related to the previous comments, I think the paper in general would benefit from a more balanced and nuanced discussion of the usefulness of both techniques, i.e. the trade-offs. For example, on line 52 the authors claim that "AR is easier to implement than DA". One could also argue that DA is "easier", or at least more modular, since it does not require changes to the model. Similarly, on line 191 the authors state that "we have no reason to suspect that other DA methods might perform better than variational DA". Without additional explanation or insights, this statement is not supported by the results in the paper. Given the wide range of DA approaches and implementations, it is not clear why this statement would hold. See also comment 5.

DA is not easier to implement than AR. Both are very easy to implement (especially compared with implementing DA in a traditional hydrological model), however it is still easier to implement AR than DA (honestly, both are trivial). We therefore do not think these tradeoffs exist in the way the reviewer is describing. We really *wanted* to find situations where DA was useful. That would connect established modeling techniques to ML approaches, and maybe open a new avenue of research.  Most importantly, as probably suspected by the reviewer, this would have made a more interesting paper. One example of this would be if it were possible to predict ahead of time which basins (with what types of characteristics) DA might be better, and develop an understanding about why. But we could not find a systematic pattern that allows for that.

4. Metrics, section 2.3: please specify what kind of forecasts you are evaluating, are these nowcasts?

Yes, nowcasting. We will add this to the methods section.

5. Methodology: results of DA typically strongly depend on how error parameters are set. Details on this aspect are provided in the appendices. We have error covariances B and R in eq.B5, which translate to alpha parameters in eq. C1. These alpha parameters are tuned during an independent validation period, with values reported in Table E1. We see that the tuned value of alpha_c (how much we trust/weight the trained model) is zero, and that alpha_y (how much we trust the real-time data) is fixed at a value of 1. If I understand it correctly, setting instead alpha_c=1 and alpha_y=0 in eq.C1 would fall back to the benchmark simulation model, i.e. not using real-time data. Why then not also tune alpha_y? Or tune some weight w=[0,1] with alpha_c=w and alpha_y=1-w? That way the DA model includes the simulation model as a special case and should never perform worse. The current results sometimes (Figures G1 and G3) show worse performance for DA than for

the benchmark simulation model. Also, are the alpha parameters the same for all basins? Why not estimate separate values for each basin?

alpha_y is set to be 1-alpha_c1. We will make this more clear in paragraph 320 (last paragraph of Appendix E).

It's a good point that separate hypertuning per basin is possible (even with a single, shared model). In an experiment that is not reported, we estimated separate hyperparameters for inference (DA) in each basin, and we also tested directing optimizer gradients toward inputs (both static and dynamic) as well as cell and hidden states in the LSTM separately per basin. There were some interesting results from this experiment, for example in some of the basins, best results came from updating only the static catchment attributes, which indicates that in these basins the catchment attribute data might be poor (this might potentially be a way to identify data errors). But the results doing this per basin vs. in batch did not change qualitatively (AR > DA on average). Coupled with the fact that AR is cheaper (does not require separate hypertuning), and the fact that in operational scenario there are often tens of thousands of gauged basins, which would make basin-specific hypertuning very expensive, our thinking was that the set of experiments reported in this paper were the most meaningful.

6. Appendix B describes variational DA and its application to LSTM. I think the math needs to be 'cleaned up' a bit for clarity:

We will fix all of the notation issues mentioned in the following comments. We also notice that there is an error in the superscript footer notation in Table E1, a missing section reference in paragraph number 315, and a duplicated word ("and") near line 320. These will be fixed in the revision.

-loss function L is written as function of model inputs x and outputs y, L(x, y), while loss is typically a function of model outputs y and corresponding observations. Where the model output depends on the unknown parameters or states for which derivatives are computed.

-Eqs. B13-B15: I don't think the gradient chains are correct, since they assume h[t] is independent of previous time slices given c[t], while the model equations B6-B11 show that there is an additional 'path' from h[t-1] to h[t]. I understand the appendix is meant to give the reader a general sense of what is happening, but you might as well write it down more correctly to avoid confusion.

-Eq. B14: the derivative on the left should be with respect to c_l

-Eq. B15: on the right we should have x and y from t-s to t instead of from 0 to t? And on the left derivative with respect to c_l[t-s], and x[t-s:t] instead of x[0:t]?

-I found it confusing that Eq. C1 switches to [t, t+s] from [t-s, t] in Eq. B15.

7. Eq. 1: what is epsilon?

8. Eq. 1: don't you want to divide by N here? Otherwise NSE values increase with N...?

9. Line 84: "is reproduced"

10. Line 199: at the time of this review, no code was provided in the linked github repository

Yes, I believe we mentioned this when we uploaded the manuscript. The repository is staged but has not been made public yet. Our intent is to make it public once we are comfortable that there are no major errors in the paper. We will do so before submitting a revised manuscript.

---

## Author Response (AR1)

**Reviewer #1**

Summary and Recommendation

Nearing et al. test how near-real time streamflow observations can effectively be used in Long Short-Term Memory (LSTM) rainfall runoff models. They compare an autoregression (AR) approach with a data assimilation (DA) approach and test, additionally, how sensitive AR is to random gaps in the data. The manuscript (MS) is easy to follow, well-structured and suits the scope of HESS. I particularly liked the comprehensive appendix in combination with a short MS. I think that this MS can be published after some minor revisions and provide only some smaller comments and questions below.

Sincerely

Ralf Loritz

Questions and comments:

- Reading the MS I would have like to see a couple of detailed results from three or four catchments where the AR or DA worked particular good or bad and what the (hydrological and ML) reason for this might be (to underpin the discussion of Appendix F and G). For instance, what could be the reason that DA and AR reduces the predictive performance of a few of your models (Fig. F2)? You state that: "*We are unsure of the reason for this, but it warrants further exploration.*" (Line 352) maybe zooming into one of the catchments could help to give a better explanation.

Thank you for the suggestion. We do agree with the notion of this comment. Indeed, We tried this extensively before submitting the manuscript and did not find a cohesive story to tell. Of course we looked at hydrographs extensively, there just isn't an interesting story to tell there.

- I find it a bit unrealistic how you added the missing data. I would assume that a broken gauging station is not working for a couple of days or maybe weeks in a row and wonder how this would alter your results (e.g. all streamflow data available for training but then two weeks or more only simulated data during testing with a closer focus on particular that period and not the entire testing period).

We did two things. First we added sequences of missing data (capped at length 10 days, since beyond this length, the model generally does not retain information from the past streamflow inputs or assimilation data. Also, we will include an "ungauged basin" analysis,

where we test both methods (DA, AR) on catchments where there is no gauge data at all during training and/or inference.

- Showing how the variance or the Shannon entropy changes of your simulations in addition to the median would be interesting (Fig.1 and Fig.3). If it remains constant, I would mention that the spread of the predictions is not affected by the data availability.

Yes, this is a strange oversight on our part. We included CDF plots in the revised manuscript.

Personal comment: Three of the seven Co-Authors have presumingly not contributed to this "technical note" as they are not mentioned in the author contribution section.

Thank you. This oversight was fixed.

**Reviewer #2**

The technical note compares two different techniques for using near-real-time streamflow observations to improve operational streamflow forecasts from LSTM rainfall-runoff models. The first technique ("autoregression", AR) adds lagged streamflow observations as predictor in the model. The second technique uses variational data assimilation (DA) to update model states within an assimilation window. The two techniques are compared on the CAMELS dataset, including experiments that artificially remove data to simulate scenarios with missing streamflow data.

The paper is generally well written, concise and to the point. The comparison between AR and DA is an interesting and novel contribution to the literature.

Comments:

1. The main conclusion is that "AR significantly out-performed the more complicated DA method" (line 195) and the authors therefore recommend against using DA (line 196). However, I feel the authors are overstating the results: differences in improved performance between AR (10%) and DA (8%) are relatively small, as also seen in Fig. 3 where the DA lines (red) and the AR lines (orange) are close.

We aren't sure how to address this. This difference is significant.

2. On line 51 it is stated that "the purpose of this paper is to provide insight into trade-offs between DA and AR". I feel the paper doesn't entirely deliver on this. Yes, the two techniques are compared across a large number of basins, but the reader doesn't get a clear sense when to use which technique. Appendix F contains a regression analysis in this direction but concludes that "we were generally unable to predict differences between the NSE scores of DA and AR". Closer inspection by a human however may lead to some insights. E.g. it could be interesting to look in more detail at extreme cases: ones where AR significantly beats DA, and vice versa. For example figure 2 shows dots in the south that are green (good) for AR and purple (bad) for DA, and vice versa.

We added nuance to much of the discussion about the comparison between these approaches. However, the bottom line is that autoregression works better in that it is generally more accurate, easier to implement, and less computationally expensive. The reviewer is correct that there is scatter in the benchmarking scatterplot. Unfortunately, as discussed in the appendix, this scatter is not predictable (as far as we have been able to discover). Because the scatter is not predictable, there is no way that we know of to determine where or when using DA might be useful. This is discussed explicitly in the paper.

3. Related to the previous comments, I think the paper in general would benefit from a more balanced and nuanced discussion of the usefulness of both techniques, i.e. the trade-offs. For example, on line 52 the authors claim that "AR is easier to implement than DA". One

could also argue that DA is "easier", or at least more modular, since it does not require changes to the model. Similarly, on line 191 the authors state that "we have no reason to suspect that other DA methods might perform better than variational DA". Without additional explanation or insights, this statement is not supported by the results in the paper. Given the wide range of DA approaches and implementations, it is not clear why this statement would hold. See also comment 5.

See previous response. Although DA in this context is easier to implement than for traditional (conceptual and process-based) hydrology models, AR is still much easier to implement and computationally less expensive during inference.

4. Metrics, section 2.3: please specify what kind of forecasts you are evaluating, are these nowcasts?

Yes, you are correct. We are doing nowcasting. We added this to the methods section.

5. Methodology: results of DA typically strongly depend on how error parameters are set. Details on this aspect are provided in the appendices. We have error covariances B and R in eq.B5, which translate to alpha parameters in eq. C1. These alpha parameters are tuned during an independent validation period, with values reported in Table E1. We see that the tuned value of alpha_c (how much we trust/weight the trained model) is zero, and that alpha_y (how much we trust the real-time data) is fixed at a value of 1. If I understand it correctly, setting instead alpha_c=1 and alpha_y=0 in eq.C1 would fall back to the benchmark simulation model, i.e. not using real-time data. Why then not also tune alpha_y? Or tune some weight w=[0,1] with alpha_c=w and alpha_y=1-w? That way the DA model includes the simulation model as a special case and should never perform worse. The current results sometimes (Figures G1 and G3) show worse performance for DA than for the benchmark simulation model. Also, are the alpha parameters the same for all basins? Why not estimate separate values for each basin?

alpha_y is set to be 1-alpha_c1. We clarified this in the revisions (last paragraph of Appendix E).

It's a good point that separate hypertuning per basin is possible (even with a single, shared model). In an experiment that is not reported, we estimated separate hyperparameters for inference (DA) in several individual basins, and we also tested directing optimizer gradients toward inputs (both static and dynamic) as well as cell and hidden states in the LSTM separately per basin. There were some interesting results from this experiment, for example in some of the basins, best results came from updating only the static catchment attributes, which indicates that in these basins the catchment attribute data might be poor (this might potentially be a way to identify data errors). But the results doing this per basin vs. in batch did not change qualitatively (AR > DA on average). Coupled with the fact that AR is cheaper (does not require separate hypertuning), and the fact that in operational scenario there are often tens of thousands of gauged basins, which would make basin-specific hypertuning

very expensive, our thinking was that the set of experiments reported in this paper are the most meaningful.

6. Appendix B describes variational DA and its application to LSTM. I think the math needs to be 'cleaned up' a bit for clarity:

We fixed all of the notation issues mentioned in the following comments.

-loss function L is written as function of model inputs x and outputs y, L(x, y), while loss is typically a function of model outputs y and corresponding observations. Where the model output depends on the unknown parameters or states for which derivatives are computed.

-Eqs. B13-B15: I don't think the gradient chains are correct, since they assume h[t] is independent of previous time slices given c[t], while the model equations B6-B11 show that there is an additional 'path' from h[t-1] to h[t]. I understand the appendix is meant to give the reader a general sense of what is happening, but you might as well write it down more correctly to avoid confusion.

-Eq. B14: the derivative on the left should be with respect to c_l

-Eq. B15: on the right we should have x and y from t-s to t instead of from 0 to t? And on the left derivative with respect to c_l[t-s], and x[t-s:t] instead of x[0:t]?

-I found it confusing that Eq. C1 switches to [t, t+s] from [t-s, t] in Eq. B15.

7. Eq. 1: what is epsilon?

8. Eq. 1: don't you want to divide by N here? Otherwise NSE values increase with N...?

9. Line 84: "is reproduced"

10. Line 199: at the time of this review, no code was provided in the linked github repository

Yes, I believe we mentioned this when we uploaded the manuscript. The repository is staged but has not been made public yet. Our intent is to make it public once we are comfortable that there are no major errors in the paper. We will do so before submitting a revised manuscript.

---

## Author Response (AR2)

I thank the authors for addressing my comments, including taking care of notational details in the equations.

The comments were extremely helpful and we are grateful that the reviewer put enough care into reading the paper especially to catch the notation issues.

I think it is a useful contribution that shows an attractive alternative to DA/inverse modeling for incorporating real-time observations of hydrological model outputs, especially for the type of models used here (LSTM).

A few remaining remarks:

1. The authors are now more nuanced in their conclusions, except for the last paragraph which makes assertions that are not supported by this study. Basically, the authors 'suspect' that their case study results are generally applicable, without giving convincing arguments or citing literature. Why not stay with the facts and simply state that more extensive and systematic benchmarking is needed to draw more general conclusions?

We agree completely, and after re-reading the last paragraph it is not good. I rewrote the last paragraph.

2. In their response, the authors mention they also did experiments where DA hyperparameters were tuned by basin. It would be interesting to mention this in the paper as a discussion item (or appendix), and give a brief quantitative indication of how much it improved things.

This is interesting, however it is a very large amount of work to make this happen. All of the experiments were re-run to account for the changes in the revisions. Incidentally, all DA and AR code was rewritten mostly from scratch during the revision because of major code refactors in the NeuralHydrology codebase between submission times. The results of the experiments were almost identical, so we are confident that there were no functional changes in the code. We would need to re-do all of these per-basin experiments in order to add them to the open source repository. This would be several days of work and weeks of run-time on GCP. Given that these results do not change the conclusions in any substantive way, we do not see this as a useful investment of time or resources.

3. Figures: some of the fonts are too small, e.g. Fig. 5

Thank you, all figures are updated with increased font size with the exception of one of the appendix figures (G1). This figure is very crowded and

Additionally, we updated the color scheme on Figure 5 to be colorblind compatible. The other figures already used colorblind-friendly color cycles.